# Solution of time-fractional gas dynamics equation using Elzaki decomposition method with Caputo-Fabrizio fractional derivative

**Maasoomah Sadaf**[1], **Zahida Perveen**[2], **Ghazala Akram**[1], **Ume Habiba**[2], **Muhammad Abbas**[3], **Homan Emadifar**[4,5]*

**1** Department of Mathematics, University of the Punjab, Quaid-e-Azam Campus, Lahore, Pakistan, **2** Department of Mathematics, Lahore Garrison University, Lahore, Pakistan, **3** Department of Mathematics, University of Sargodha, Sargodha, Pakistan, **4** Department of Mathematics, Hamedan Branch, Islamic Azad University, Hamedan, Iran, **5** MEU Research Unit, Middle East University, Amman, Jordan

* homan_emadi@yahoo.com

**Data Availability Statement:** All relevant data is within the paper.

**Funding:** The author(s) received no specific funding for this work.

## Abstract

In this article, Elzaki decomposition method (EDM) has been applied to approximate the analytical solution of the time-fractional gas-dynamics equation. The time-fractional derivative is used in the Caputo-Fabrizio sense. The proposed method is implemented on homogenous and non-homogenous cases of the time-fractional gas-dynamics equation. A comparison between the exact and approximate solutions is also provided to show the validity and accuracy of the technique. A graphical representation of all the retrieved solutions is shown for different values of the fractional parameter. The time development of all solutions is also represented in 2D graphs. The obtained results may help understand the physical systems governed by the gas-dynamics equation.

## 1 Introduction

The physical laws of energy conservation, momentum conservation and mass conservation are defined by the mathematical representation of gas-dynamics equations. Gas dynamics is a branch of fluid dynamics that studies gas motion and its effect on physical construction. The study of gas-dynamics has a number of useful applications in various problems of science and engineering, such as; choked flows in nozzles and pipes, shock waves around aircrafts, aerodynamic heating on atmospheric reentry car and others. Recently, many researchers have investigated gas-dynamics equation in various studies owing to its significance in different physical phenomena. The different techniques utilized to study the gas-dynamics equation include finite-difference method [1], q-Homotopy analysis method [2], a combination of Laplace transform and homotopy perturbation method [3], Elzaki transform homotopy perturbation method [4], fractional homotopy analysis transform method [5], homotopy-perturbation method [6], quadratic B-spline Galerkin method [7], combination of integral and projected differential transform method [8] and fractional variational iteration method [9].

**Competing interests:** The authors have declared that no competing interests exist.

The nonlinear fractional order gas-dynamics equation is considered, as [10]

$$\frac{\partial^{\beta} w}{\partial t^{\beta}} + w\frac{\partial w}{\partial \psi} - w(1-w) = 0, \ t \in R, \ 0 < \beta \leq 1. \tag{1}$$

The initial condition is $w(\psi, 0) = k(x)$, where $\beta$ is a parameter that describes the fractional order of derivative. When $\beta = 1$, Eq (11) reduces to the classical integer order gas-dynamics equation. The fractional gas-dynamics equation has been examined using different approaches. Das and Kumar [11] utilized differential transform method to solve the problem considering the fractional derivative in Caputo sense. Iqbal et al. [10] presented an iterative technique using Caputo fractional derivative to solve the fractional gas-dynamics equation. Iyiola [2] determined the solution of fractional gas-dynamics equation using q-homotopy analysis method with Caputo fractional derivative.

The time-fractional differential operators are more generalized than the integer order differential equations appearing in classical calculus. The fractional calculus has become increasingly popular over the last few years. The basic notions of fractional derivative are introduced by Caputo and Riemann-Liouville, which involve the singular kernal [12]

$$k(t,s) = \frac{(t-s)^{-\beta}}{\Gamma(1-\beta)}, \ 0 < \beta < 1. \tag{2}$$

However, Caputo and Fabrizio noted in [13] that the modeling of many phenomena in physics cannot be well-modeled using Caputo and Riemann-Liouville fractional derivatives. In order to solve this problem, Caputo and Fabrizio introduced a novel definition of fractional derivative with a non-singular kernal [13]

$$k(t,s) = e^{\frac{-\beta(t-s)}{1-\beta}}, \ 0 < \beta < 1. \tag{3}$$

Presently, Caputo-Fabrizio derivative is one of the most commonly used definition of time-fractional derivative which is employed for the solution of many mathematical problems in engineering sciences. Caputo-Fabrizio derivative has been successfully used in the study of general form of Walter's-B fluid model [14], a new dynamical model of hepatitis E [15], a new fractional differential model for COVID-19 transmission [16], mathematical modeling of human liver [17] and others. The Caputo-Fabrizio derivative has been used to solve fractional Sharma-Tasso-Olver-Burgers equation and (2+ 1)-dimensional mKdV equation [18, 19].

The main objective of this manuscript is to propose a novel analytical technique for the solution of time-fractional gas-dynamics equation using the Caputo-Fabrizio derivative. The proposed technique utilizes the Elzaki transform and Caputo-Fabrizio fractional derivative along with the Adomian polynomials to construct the approximate analytical solution of the time-fractional gas-dynamics equation. Two numerical applications are presented to illustrate the proposed method for homogeneous and in-homogeneous case. The change in the solution under the influence of fractional parameter is observed through numerical and graphical observations.

Elzaki transform was first introduced by Tarig Elzaki [20]. Adomian decomposition method [21] is a well-known mathematical technique to solve the nonlinear partial differential equations. Many researchers have applied Elzaki transform combined with the Adomian decomposition method on a variety of problems to find their solution such as; epidemic model [22], fifth-order Korteweg-De Kries equations [23], sine-Gordon equation [24], linear and nonlinear Schrödinger equation [25] and nonlinear equation for water inflation in unsaturated soil [26].

## 2 Fundamental definitions and results

**Definition 1** [27] Let $0 < \beta < 1$ and $w$ be a continuously differentiable function. The CF fractional derivative of $w$ of order $\beta$ is given by

$$D_t^{\beta} w(t) = \frac{1}{1-\beta} \int_0^t \exp\left(-\frac{\beta(t-s)}{1-\beta}\right) w'(s) ds. \tag{4}$$

**Definition 2** [28] The Elzaki transform is defined over the set of functions

$$B = \{g(t) / \exists M, k_1, k_2 > 0, |g(t)| < M \exp\left(\frac{|t|}{k_j}\right), if\ t \in (-1)^j \times [0, \infty)\}, \tag{5}$$

by the following integral

$$E|g(t)| = T(s) = s \int_0^{\infty} g(t) \exp\left(\frac{-t}{s}\right) dt, t > 0, \tag{6}$$

where $s$ is the factor of variable $t$.

Elzaki transform exhibits the following useful properties [28].

1. Convolution property

$$E[w(t) * g(t)] = \frac{1}{s} E[w(t)] E[g(t)]. \tag{7}$$

2. Differentiation property

If $w^{(m)}(t)$ is the $m$-th time-derivative of the function $w(t) \in B$ then its Elzaki transform is given by

$$E[w^{(m)}(t)] = \frac{1}{s^m} T(s) - \sum_{n=0}^{m-1} s^{2-m+n} w^{(n)}(0). \tag{8}$$

**Theorem 3** [29] The Elzaki transform of the CF fractional derivative can be expressed, as

$$E\left[D_t^{\beta}(w(t))\right] = \frac{s\left(\dfrac{W(s)}{s} - sw(0)\right)}{1 - \beta(1-s)}. \tag{9}$$

In general,

$$E\left[D_t^{m+\beta}(w(t))\right] = \frac{s\left(\dfrac{W(s)}{s^{m+1}} - \sum_{n=0}^{m} s^{1-m+n} w^{(n)}(0)\right)}{1 - \beta(1-s)}. \tag{10}$$

## 3 Description of methodology

The fractional differential equation for $w(\psi, t)$ is considered, as

$$D_t^{\beta} w(\psi, t) + Pw(\psi, t) + Qw(\psi, t) = k(\psi, t),\ r \in N,\ r - 1 < \beta \leq r. \tag{11}$$

where $P$ and $Q$ are nonlinear and linear terms and $D_t^{\beta}$ denotes the time-fractional Caputo-Fabrizio differential operator. The initial condition is considered in accordance with [2, 10, 11], as

$$w(\psi, 0) = v_r(\psi). \tag{12}$$

Applying Elzaki transform with fractional order Caputo-Fabrizio derivative and using Theorem 3, Eq (11) implies

$$\frac{s\left(\dfrac{W(\psi, s)}{s} - sw(\psi, 0)\right)}{1 - \beta(1 - s)} = E[k(\psi, t) - Pw(\psi, t) - Qw(\psi, t)]. \tag{13}$$

Using the initial condition given by Eq (12), the following relation is obtained.

$$\frac{W(\psi, s) - s^2 v_r(\psi)}{1 - \beta(1 - s)} = E[k(\psi, t) - Pw(\psi, t) - Qw(\psi, t)], \tag{14}$$

or

$$W(\psi, s) = s^2 v_r(\psi) + [1 - \beta(1 - s)]E[k(\psi, t) - Pw(\psi, t) - Qw(\psi, t)]. \tag{15}$$

Applying inverse Elzaki transform on both sides of Eq (15),

$$w(\psi, t) = E^{-1}[s^2 v_r(\psi) + [1 - \beta(1 - s)]E[k(\psi, t) - Pw(\psi, t) - Qw(\psi, t)]], \tag{16}$$

$$w(\psi, t) = v_r(\psi) + E^{-1}[1 - \beta(1 - s)]E[k(\psi, t) - Pw(\psi, t) - Qw(\psi, t)]. \tag{17}$$

Using the Adomian decomposition technique [21], the series expansion of the solution is assumed, as

$$w(\psi, t) = \sum_{n=0}^{\infty} w_n(\psi, t). \tag{18}$$

Using the decomposition defined by Eq (18), Eq (17) can be rewritten in the following form.

$$\sum_{n=0}^{\infty} w_n(\psi, t) = v_r(\psi) + E^{-1}[1 - \beta(1 - s)]E[k(\psi, t)$$
$$- P\sum_{n=0}^{\infty} w_n(\psi, t) - Q\sum_{n=0}^{\infty} w_n(\psi, t)]. \tag{19}$$

The following recursive relation is obtained by the term by term comparison on both sides of Eq (19).

$$w_0(\psi, t) = v_r(\psi), \tag{20}$$

$$w_{n+1}(\psi, t) = E^{-1}[1 - \beta(1 - s)]E[k(\psi, t) - Pw_n(\psi, t) - Qw_n(\psi, t)], \ n \geq 0. \tag{21}$$

The approximate analytical solution can be determined, as

$$w(\psi, t) \approx w_0(\psi, t) + w_1(\psi, t) + w_2(\psi, t) + \ldots w_m(\psi, t), \ m = 0, 1, 2 \ldots \tag{22}$$

## 4 Applications

**Example 3.1** *Consider the fractional order gas-dynamics equation of the form*

$$D_t^\beta w + \frac{1}{2}(w^2)_\psi = w - w^2, \ 0 < \beta \leq 1, \ t > 0, \tag{23}$$

*with the initial condition*

$$w(\psi, 0) = e^{-\psi}. \tag{24}$$

*Applying Elzaki transform on both sides of* Eq (23) *with fractional order Caputo-Fabrizio derivative and using* Eq (34), *the following equation is obtained.*

$$W(\psi, s) = s^2 e^{-\psi} + [1 - \beta(1-s)]\left[E(w - w^2 - \frac{1}{2}(w^2)_\psi)\right]. \tag{25}$$

*Applying inverse Elzaki transform on* (25), *the resulting equation becomes.*

$$w(\psi, t) = e^{-\psi} + E^{-1}[1 - \beta(1-s)]\left[E(w - w^2 - \frac{1}{2}(w^2)_\psi)\right]. \tag{26}$$

*The recursive relation given by* Eqs (20) *and* (22) *can be expressed, as*

$$w_0(\psi, 0) = e^{-\psi}. \tag{27}$$

$$
\sum_{n=0}^{\infty} w_{n+1}(\psi, t) = e^{-\psi} + E^{-1}[1 - \beta(1-s)[E(\sum_{n=0}^{\infty} w_n(\psi, t) \\
- \sum_{n=0}^{\infty} w_n^2(\psi, t) - \frac{1}{2}(\sum_{n=0}^{\infty} w_n^2(\psi, t))_\psi)]]. \tag{28}
$$

*The successive terms are determined, as follows:*

$$
\begin{aligned}
w_1(\psi, t) &= E^{-1}\left[\{1 - \beta(1-s)\}\left[E(w_0 - w_0^2 - \frac{1}{2}(w_0^2)_\psi)\right]\right] \\
&= e^{-\psi} E^{-1}\{1 - \beta(1-s)\}s^2 \\
&= e^{-\psi}(1 - \beta + \beta t),
\end{aligned} \tag{29}
$$

$$
\begin{aligned}
w_2(\psi, t) &= E^{-1}[\{1 - \beta(1-s)\}[E(e^{-\psi}(1 - \beta + \beta t) - (e^{-\psi}(1 - \beta + \beta t))^2 \\
&\quad - \frac{1}{2}(e^{-2\psi}(1 - \beta + \beta t)^2)_\psi)]] \\
&= e^{-\psi} E^{-1}[\{1 - \beta(1-s)\}[(s^2 - \beta s^2 + \beta s^3)]] \\
&= e^{-\psi}[(1 - \beta)^2 + \beta t(2 - 2\beta + \beta t)].
\end{aligned} \tag{30}
$$

*The solution is expressed, as:*

$$w(\psi, t) = e^{-\psi} + e^{-\psi}(1 - \beta + \beta t) + e^{-\psi}[(1 - \beta)^2 + \beta t(2 - 2\beta + \beta t)] + \ldots. \tag{31}$$

*The exact solution of the problem at* $\beta = 1$ *available in literature* [2, 10, 11], *as*

$$w(\psi, t) = \exp^{-\psi + t}. \tag{32}$$

**Example 3.2** *Consider the nonlinear non homogenous fractional order gas-dynamics equation*

$$D_t^\beta w + w w_\psi - w(1 - w) + e^{-\psi + t} = 0, \ 0 < \beta \le 1, \tag{33}$$

*with the initial condition*

$$w(\psi, 0) = 1 - e^{-\psi}. \tag{34}$$

*Applying Elzaki transform on both sides of* Eq (33) *with fractional order Caputo-Fabrizio derivative and using* Eq (34), *the following relation relation is obtained.*

$$W(\psi, s) = s^2(1 - e^{-\psi}) + \{1 - \beta(1 - s)\}E[w(1 - w) - e^{-\psi+t} - ww_\psi]. \tag{35}$$

*Application of inverse Elzaki transform on both sides of* Eq (35) *implies*

$$w(\psi, t) = (1 - e^{-\psi}) + E^{-1}[\{1 - \beta(1 - s)\}E[w(1 - w) - e^{-\psi+t} - ww_\psi]]. \tag{36}$$

*The recursive relation given by* Eqs (20) *and* (21) *takes the following form.*

$$w_0(\psi, t) = 1 - e^{-\psi}, \tag{37}$$

$$
\sum_{n=0}^{\infty} w_{n+1}(\psi, t) = (1 - e^{-\psi}) + E^{-1}\left[\{1 - \beta(1 - s)\}E\left[\sum_{n=0}^{\infty} w_n(\psi, t)\left(1 - \sum_{n=0}^{\infty} w_n(\psi, t)\right)\right.\right.
$$
$$
\left.\left. -e^{-\psi+t} - \sum_{n=0}^{\infty} w_n(\psi, t)\frac{\partial}{\partial\psi}\sum_{n=0}^{\infty} w_n(\psi, t)\right]\right]. \tag{38}
$$

**Table 1. Approximated solution of Example 3.1 at $t = 0.01$.**

| $\psi/\beta$ | 0.5 | 0.6 | 0.7 | 0.8 | 0.9 | 1 |
|---|---|---|---|---|---|---|
| 0.1 | 1.592536477 | 1.421351190 | 1.267902527 | 1.132190487 | 1.014215069 | 0.913976275 |
| 0.2 | 1.440986593 | 1.286091741 | 1.147245649 | 1.024448317 | 0.917699945 | 0.826999933 |
| 0.3 | 1.303858588 | 1.163703930 | 1.038070790 | 0.926959170 | 0.830636906 | 0.748300484 |
| 0.4 | 1.179780039 | 1.052962859 | 0.939285294 | 0.838743420 | 0.751349003 | 0.677090278 |
| 0.5 | 1.067509124 | 0.952760195 | 0.849900480 | 0.758929979 | 0.679848692 | 0.612656193 |
| 0.6 | 0.965922199 | 0.862093075 | 0.769021756 | 0.686708243 | 0.615152535 | 0.554354633 |
| 0.7 | 0.874002549 | 0.780054072 | 0.695839660 | 0.621359313 | 0.556613031 | 0.501600815 |
| 0.8 | 0.790830210 | 0.705822112 | 0.629621761 | 0.562229156 | 0.503644298 | 0.453867186 |
| 0.9 | 0.715572765 | 0.638654258 | 0.569705329 | 0.508725978 | 0.455716206 | 0.410676013 |
| 1 | 0.647477013 | 0.577878269 | 0.515490699 | 0.460314301 | 0.412349075 | 0.371595023 |

**Table 2. Approximated solution of Example 3.1 at $t = 0.005$.**

| $\psi/\beta$ | 0.5 | 0.6 | 0.7 | 0.8 | 0.9 | 1 |
|---|---|---|---|---|---|---|
| 0.1 | 1.587995323 | 1.416440637 | 1.262802184 | 1.270799653 | 1.009273979 | 0.909384226 |
| 0.2 | 1.436877588 | 1.281648489 | 1.142630668 | 1.019824125 | 0.913228612 | 0.822844875 |
| 0.3 | 1.300140607 | 1.159683510 | 1.033894983 | 0.922775028 | 0.826323644 | 0.744540832 |
| 0.4 | 1.176415870 | 1.049325032 | 0.935506867 | 0.834961374 | 0.747688553 | 0.673688404 |
| 0.5 | 1.064465098 | 0.949468553 | 0.846481618 | 0.755054294 | 0.676536580 | 0.609578476 |
| 0.6 | 0.963167851 | 0.859114674 | 0.765928242 | 0.683608554 | 0.612155612 | 0.551694145 |
| 0.7 | 0.871510311 | 0.777359103 | 0.693040533 | 0.618554599 | 0.553901303 | 0.499080644 |
| 0.8 | 0.788575140 | 0.703383604 | 0.627089006 | 0.559691346 | 0.501190625 | 0.451586842 |
| 0.9 | 0.713532293 | 0.636447804 | 0.567413597 | 0.506429673 | 0.453496031 | 0.408612672 |
| 1 | 0.645630718 | 0.575881788 | 0.513417054 | 0.458236517 | 0.410340178 | 0.369728035 |

*The next term in the series can be computed, as*

$$
\begin{aligned}
w_1(\psi, t) &= E^{-1}\left[\{1 - \beta(1-s)\}E\left[w_0(1 - w_0) - e^{-\psi+t} - w_0\frac{\partial w_0}{\partial \psi}\right]\right] \\
&= -e^{-\psi}(e^t - \beta).
\end{aligned}
\tag{39}
$$

**Table 3. Comparison between exact and approximated solutions of Example 3.1 at $\beta = 1$.**

| $\psi/t$ | Exact at 0.01 | Approximated at 0.01 | Exact at 0.005 | Approximated at 0.005 |
|---|---|---|---|---|
| 0.1 | 0.913931185 | 0.913976275 | 0.909372934 | 0.909384226 |
| 0.2 | 0.826959133 | 0.826999933 | 0.822834658 | 0.822844875 |
| 0.3 | 0.748263567 | 0.748300484 | 0.744531587 | 0.744540832 |
| 0.4 | 0.677056874 | 0.677090278 | 0.673680039 | 0.673688404 |
| 0.5 | 0.612626394 | 0.612656619 | 0.609570907 | 0.609578479 |
| 0.6 | 0.554327284 | 0.554354633 | 0.551562566 | 0.551569414 |
| 0.7 | 0.501576069 | 0.501600815 | 0.499074447 | 0.499080644 |
| 0.8 | 0.453844795 | 0.453867186 | 0.451581284 | 0.451586842 |
| 0.9 | 0.410655752 | 0.410676013 | 0.408607598 | 0.408612672 |
| 1 | 0.371576691 | 0.371595023 | 0.369723444 | 0.369728035 |

**Table 4. Approximate solution of Example 3.1 at $t = 0.01$.**

| $\psi/\beta$ | 0.5 | 0.6 | 0.7 | 0.8 | 0.9 | 1 |
|---|---|---|---|---|---|---|
| 0.1 | −0.36634989 | −0.27586615 | −0.18538241 | −0.09489668 | −0.00441492 | 0.086068814 |
| 0.2 | −0.23632451 | −0.15445143 | −0.07257835 | 0.929471544 | 0.092116779 | 0.173040866 |
| 0.3 | −0.11867267 | −0.04459085 | 0.029490966 | 0.103572788 | 0.177654610 | 0.251736432 |
| 0.4 | −0.01221698 | 0.548151070 | 0.121847111 | 0.188879116 | 0.255911120 | 0.322943125 |
| 0.5 | 0.084108275 | 0.144713419 | 0.205414407 | 0.266067473 | 0.326720539 | 0.387373605 |
| 0.6 | 0.017126689 | 0.226148060 | 0.281029224 | 0.335910388 | 0.390791551 | 0.445672715 |
| 0.7 | 0.250131279 | 0.299789809 | 0.349448339 | 0.399106870 | 0.448765400 | 0.498423930 |
| 0.8 | 0.321490722 | 0.366423619 | 0.411356515 | 0.456289411 | 0.548687281 | 0.546155204 |
| 0.9 | 0.386059417 | 0.426716383 | 0.467373349 | 0.508030315 | 0.548687281 | 0.599344247 |
| 1 | 0.444483588 | 0.481271532 | 0.518059476 | 0.554847420 | 0.591635364 | 0.628423308 |

**Table 5. Approximated solution of Example 3.2 at $t = 0.005$.**

| $\psi/\beta$ | 0.5 | 0.6 | 0.7 | 0.8 | 0.9 | 1 |
|---|---|---|---|---|---|---|
| 0.1 | −0.36791643 | −0.27130790 | −0.18082415 | −0.09034041 | 0.000143323 | 0.090627065 |
| 0.2 | −0.24322000 | −0.15032695 | −0.06845388 | 0.013419132 | 0.095292266 | 0.177165341 |
| 0.3 | −0.11494069 | −0.04085887 | 0.033222946 | 0.107304768 | 0.181386590 | 0.255468412 |
| 0.4 | −0.00884006 | 0.058191942 | 0.125223946 | 0.192255951 | 0.259287956 | 0.326319960 |
| 0.5 | 0.087163762 | 0.147816828 | 0.208469894 | 0.269122960 | 0.329776026 | 0.390429092 |
| 0.6 | 0.174031616 | 0.228912779 | 0.283793943 | 0.338675106 | 0.393556270 | 0.448437434 |
| 0.7 | 0.252632900 | 0.302291430 | 0.351949960 | 0.401608491 | 0.451267052 | 0.500092555 |
| 0.8 | 0.323754283 | 0.368687179 | 0.413620075 | 0.458552972 | 0.503485868 | 0.548418765 |
| 0.9 | 0.388107571 | 0.428764537 | 0.469421503 | 0.510078469 | 0.550735435 | 0.591392401 |
| 1 | 0.446336834 | 0.483124778 | 0.519912723 | 0.556700672 | 0.593488611 | 0.630276555 |

**Table 6. Comparison between exact and approximated solutions of Example 3.2 at $\beta = 1$.**

| $\psi t$ | Exact at 0.01 | Approximated at 0.01 | Exact at 0.005 | Approximated at 0.005 |
|---|---|---|---|---|
| 0.1 | 0.086068814 | 0.086068814 | 0.090627065 | 0.090627065 |
| 0.2 | 0.173040866 | 0.173040866 | 0.177165314 | 0.177165314 |
| 0.3 | 0.251736432 | 0.251736432 | 0.255468412 | 0.255468412 |
| 0.4 | 0.322943125 | 0.322943125 | 0.326319960 | 0.326319960 |
| 0.5 | 0.387373605 | 0.387373605 | 0.390429092 | 0.390429092 |
| 0.6 | 0.445672715 | 0.445672715 | 0.448437434 | 0.448437434 |
| 0.7 | 0.498423930 | 0.498423930 | 0.500925552 | 0.500925552 |
| 0.8 | 0.546155204 | 0.546155204 | 0.548418765 | 0.548418765 |
| 0.9 | 0.589344247 | 0.589344247 | 0.591392401 | 0.591392401 |
| 1 | 0.628423308 | 0.628423308 | 0.630276555 | 0.630276555 |

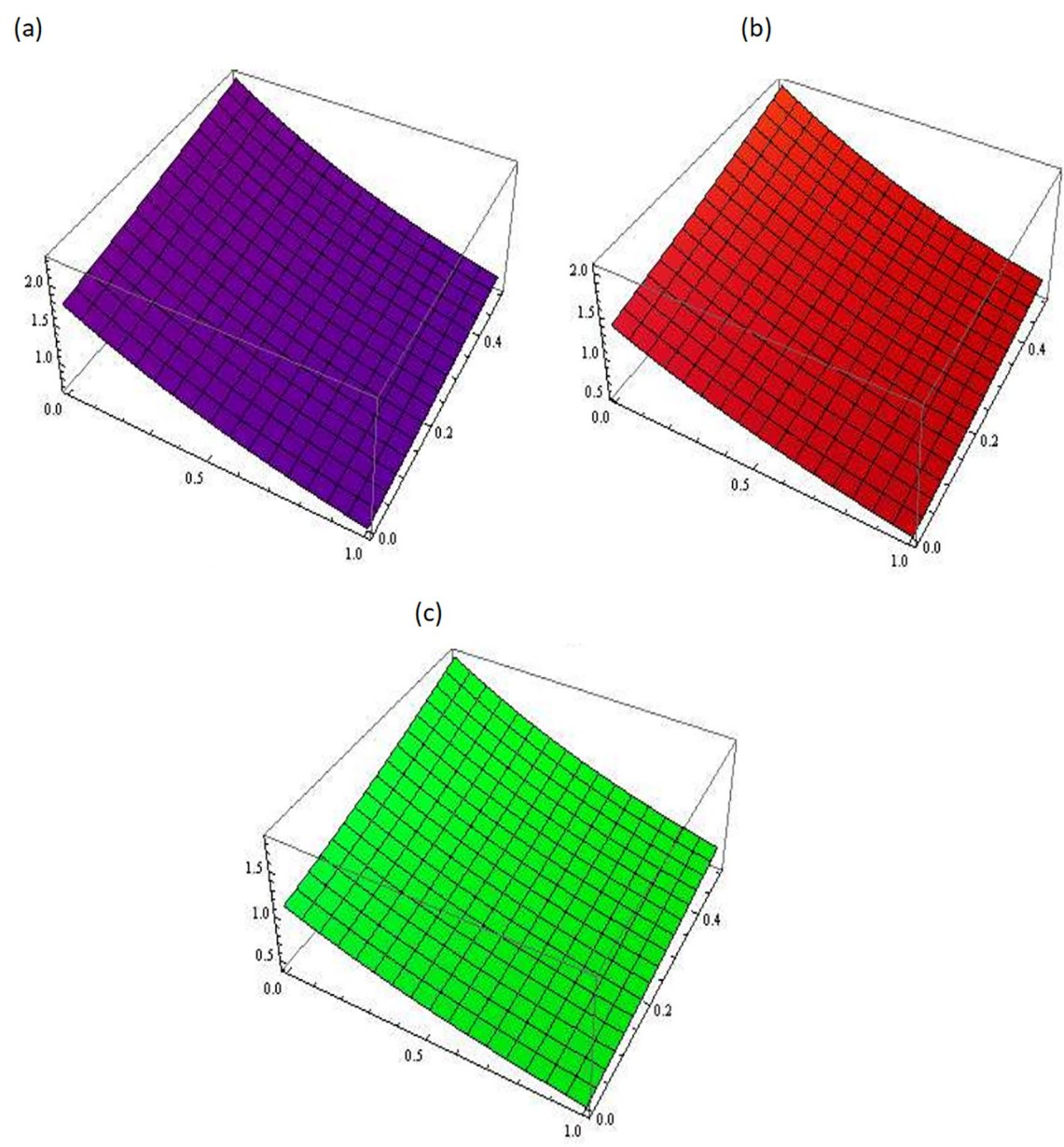

**Fig 1.** Approximated solution of Example 3.1 at different values of $\beta$, (a): $\beta = 0.5$, (b): $\beta = 0.7$, (c): $\beta = 0.9$.

*The solution can be expressed, as*

$$w(\psi, t) = 1 - e^{-\psi} - e^{-\psi}(e^t - \beta) + \dots \tag{40}$$

*The exact solution of the problem for $\beta = 1$ is available in literature* [10]*, as*

$$w(\psi, t) = 1 - \exp^{-\psi + t}. \tag{41}$$

(a)
(b)

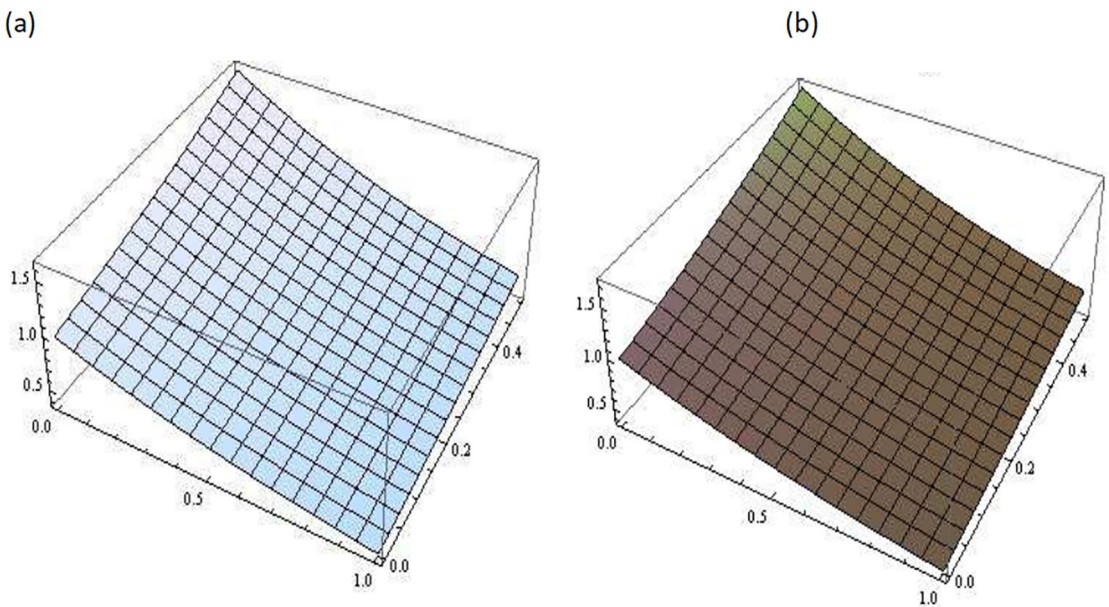

**Fig 2.** Comparison of exact and approximate solutions of Example 3.1 at $\beta = 1$, (a): Exact Solution, (b): Approximated Solution.

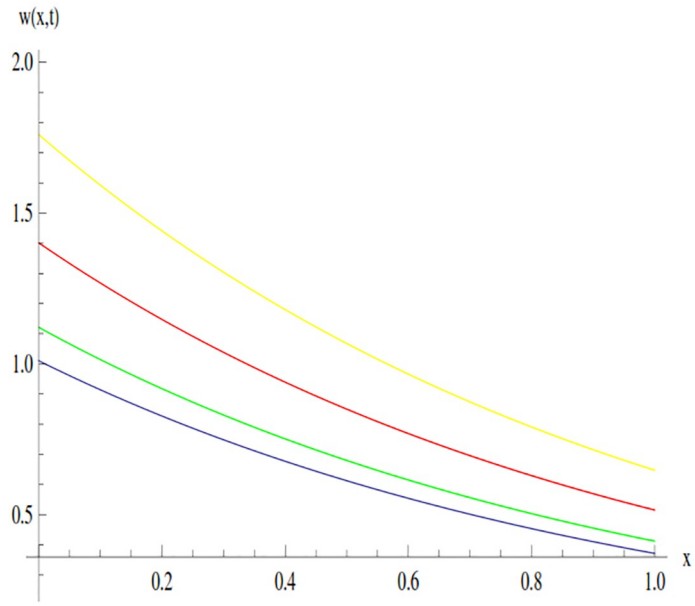

**Fig 3. Variation in the solution of Example 3.1 for different values of $\beta$ at $t = 0.01$.**

## 5 Results and simulations

The approximated numerical values for Example 3.1 are summarized in Tables 1 and 2. Comparison between the exact and approximated solution provided in Table 3 show the accuracy of the determined solution. The approximated mathematical calculations for Example 3.2 are summarized in Tables 4 and 5. The efficacy of the proposed method is established through comparison between the exact and approximated solutions as shown in Table 6.

The physical behavior of the solution of gas-dynamics equation gained by using the proposed methodology involving the Elzaki transform with the Caputo-Fabrizio fractional differential operator is observed through graphs. The solution obtained by the presented method is in series form and its value changes with the change in the fractional order $\beta$ of derivative.

Fig 1 describes the three-dimensional graph of Example 3.1 for different values of fractional parameter *i.e.* $\beta = 0.5, 0.7, 0.9$. It demonstrates the variation in the numerical results obtained for different values of $\beta$. Fig 2 shows the comparison between the exact and the approximate solutions of Example 3.1 at $\beta = 1$. This comparison shows a strong agreement between the obtained solution and exact solution.

The behavior of the solution of Example 3.1 at time 0.01 and 0.005 is shown through line graphs presented in Figs 3 and 4. Different colors are used to depict the line graph at different values of $\beta$ to show the comparison. The yellow, red, green and blue lines indicated the plots of solution at $\beta = 0.5, \beta = 0.7, \beta = 0.9$ and $\beta = 1$, respectively.

The effect of fractional order $\beta$ on the solution of Example 3.2 is graphically illustrated in Figs 5–8. The graph in Fig 5 shows the physical behavior of the obtained solution using the presented technique at $\beta = 0.5, 0.7$ and $\beta = 0.9$. Fig 6 shows the comparison between the exact and the approximate solutions of Example 3.2 at $\beta = 1$. This comparison shows a strong connection among the solution and the exact solution. Figs 7 and 8 present the line plots of the solution of Example 3.2 for $\beta = 0.5, 0.7, 0.9$ at $t = 0.01$ and $t = 0.005$ respectively.

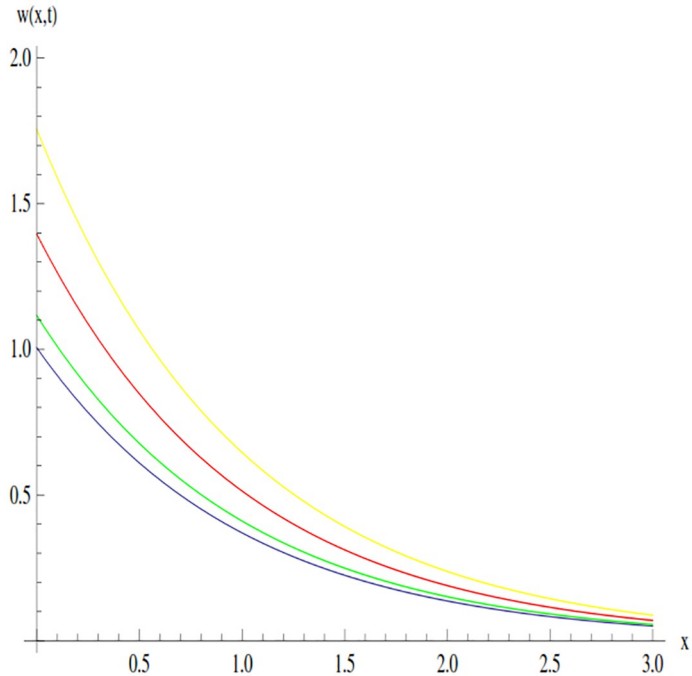

**Fig 4. Variation in the solution of Example 3.1 for different values of $\beta$ at $t = 0.005$.**

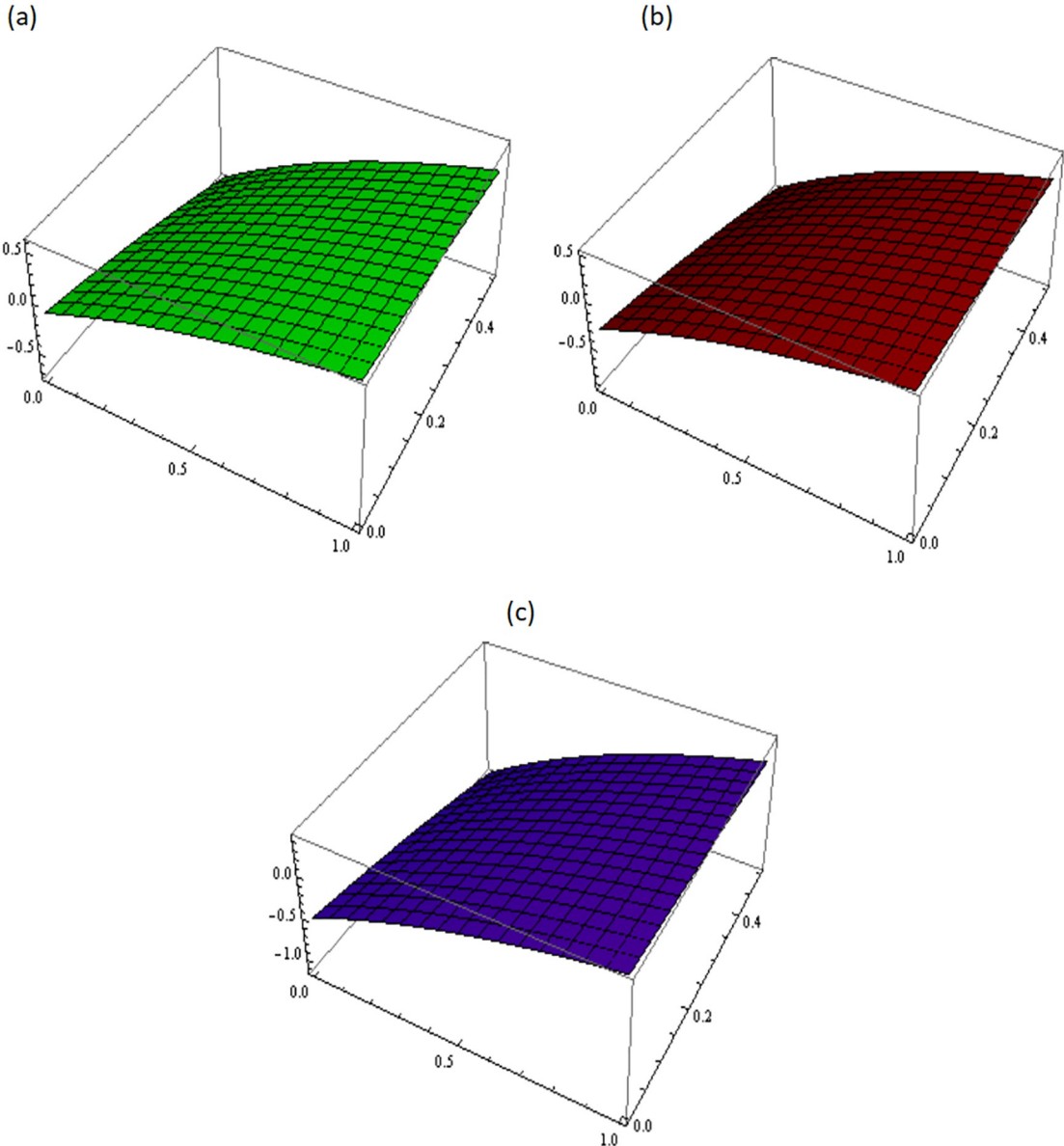

**Fig 5.** Approximated solution of Example 3.2 at different values of $\beta$, (a): $\beta = 0.5$, (b): $\beta = 0.7$, (c): $\beta = 0.9$.

## 6 Discussion of the results

The numerical illustration of the proposed method is presented using two examples in Section 3. These two examples have been previously studied with different fractional time-derivatives and their exact solutions are known at $\beta = 1$ [2, 10, 11]. Thus, the selected numerical examples allow a comparison of the obtained results with the results available in literature. It is worth mentioning that the CF derivative is utilized for the first time to solve the fractional order gas-dynamics equation in this work. On comparing the graphs for different values of the fractional order $\beta$ with the graphs given in [2, 10, 11], it is observed that the CF derivative employed in the proposed Elzaki Adomian decomposition method provides results with good accuracy

(a)          (b)

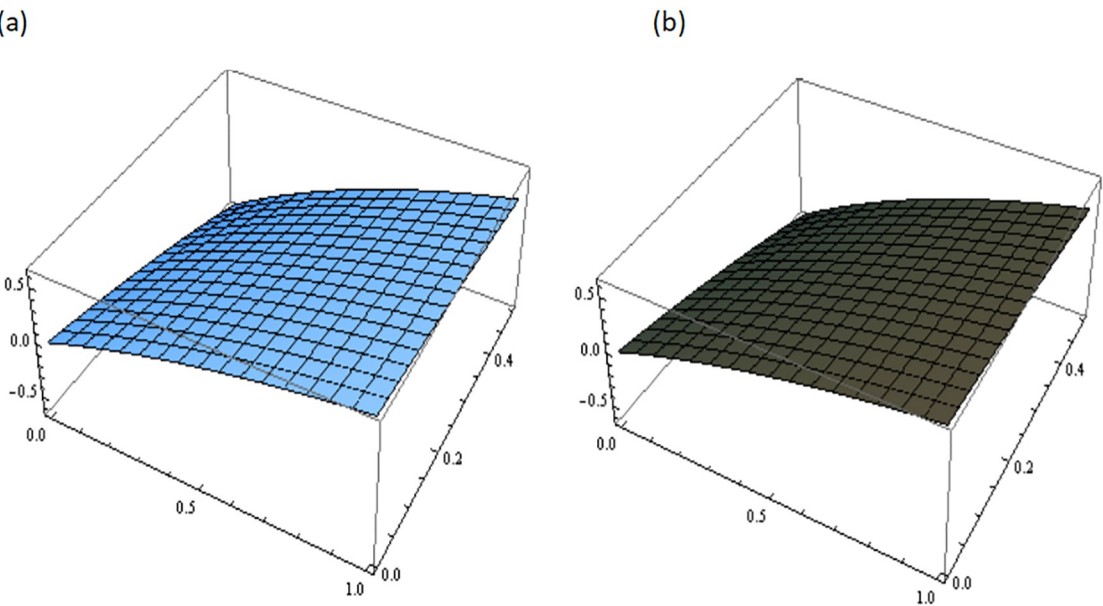

**Fig 6.** Comparison of exact and approximate solutions of Example 3.2 at $\beta = 1$, (a): Exact Solution, (b): Approximated Solution.

with only a small number of terms calculated in the power series solution. The obtained results are compared with the exact solutions at $\beta = 1$ which are given in [2, 10, 11] and the results are summarized Tables 3 and 6, Figs 2 and 6 which confirm the accuracy of the obtained results.

Based on the above comparisons, it can be concluded that the proposed technique can be effectively applied to determine the solution of homogeneous and non-homogeneous gas-dynamics equations with given initial conditions with high accuracy. Since the theory of

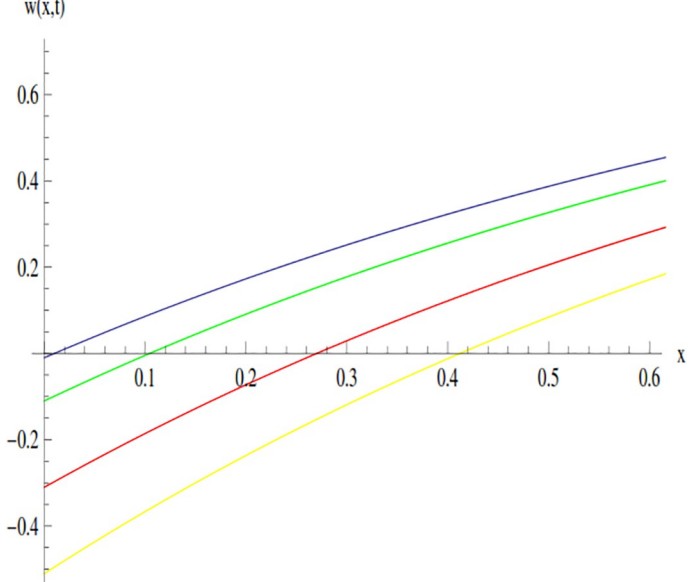

**Fig 7. Variation in the solution of Example 3.2 for different values of $\beta$ at $t = 0.01$.**

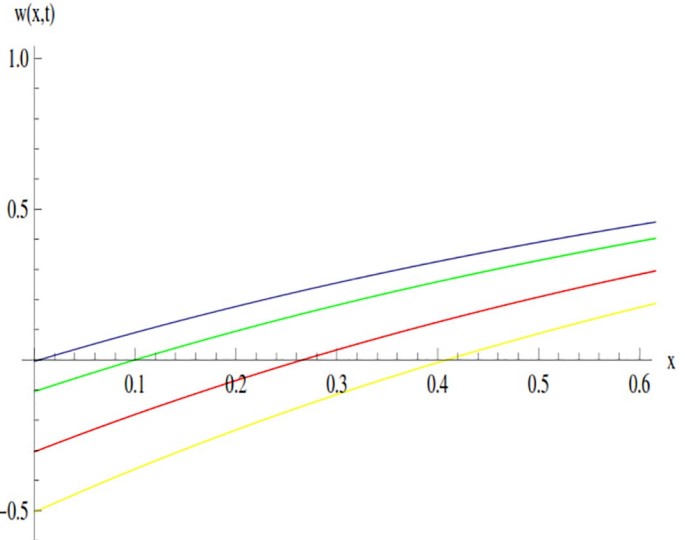

**Fig 8. Variation in the solution of Example 3.2 for different values of *β* at *t* = 0.005.**

fractional calculus and fractional order models is still evolving, the physical applications of the use of fractional derivative are yet to be fully explored. However, we can observe that the solution surface for the gas-dynamics equation continuously changes for increasing value of fractional order *β*. Ultimately, the solution surface at *β* = 1 coincides with the solution of the classical integer order gas-dynamics equation which confirms that CF derivative is indeed a generalization of the classical integer order derivative.

## 7 Conclusion

The fractional gas-dynamics equation arises in the study of gas motion and its effect on physical construction. In this work, a novel analytical method is proposed to retrieve the analytical approximate solutions of the time-fractional gas-dynamics equation with a fractional temporal operator defined in the Caputo-Fabrizio sense. The proposed method employs the concepts of the Elzaki transform along with the Adomian decomposition. The presented method is demonstrated with the help of two numerical applications. Numerical and graphical observations for the applications are also provided which show the efficiency of the proposed method for accurate solutions of fractional-order homogenous and non-homogenous gas-dynamics equations. From Tables 1 and 2 for Example 3.1 and Tables 4 and 5 for Example 3.2, it is evident that the solution of the gas-dynamics equation varies with increasing value of *β*. The accuracy of the obtained solutions is established through the comparison of the obtained solutions at *β* = 1 with the exact solutions available in the literature as shown in Tables 3 and 6. The obtained results are also explained through the graphical simulations presented in Figs 1–8. The reported results establish the accuracy of the developed mathematical technique. Moreover, the obtained solution may help to explore many problems related to the gas-dynamics equation.

## Acknowledgments

The authors are grateful to anonymous referees for their valuable suggestions, which significantly improved this manuscript.

## Author Contributions

**Conceptualization:** Zahida Perveen.

**Data curation:** Ghazala Akram.

**Investigation:** Ume Habiba.

**Resources:** Muhammad Abbas.

**Software:** Muhammad Abbas.

**Writing – original draft:** Maasoomah Sadaf.

**Writing – review & editing:** Homan Emadifar.

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
