## [Decision Letter · Decision Letter 0]

7 Nov 2023

PONE-D-23-34314Solution of time-fractional gas dynamics equation using Elzaki decomposition method with Caputo-Fabrizio fractional derivativePLOS ONE

Dear Dr. Emadifar,

Thank you for submitting your manuscript to PLOS ONE. After careful consideration, we feel that it has merit but does not fully meet PLOS ONE’s publication criteria as it currently stands. Therefore, we invite you to submit a revised version of the manuscript that addresses the points raised during the review process.

- Authors must respond to reviewers' comments.

- Authors are not obligated to cite suggested references unless these references are relevant.

- Authors must polish the English language and place punctuation devices in the appropriate places. Best regards

Mohammed S. Abdo

Academic Editor

We look forward to receiving your revised manuscript.

Kind regards,

Mohammed S. Abdo

Academic Editor

PLOS ONE

Journal Requirements:

Additional Editor Comments:

Dear Authors,

we have received the reports from our advisors on your manuscript,

"Solution of time-fractional gas dynamics equation using Elzaki decomposition method with Caputo-Fabrizio fractional derivative", which you submitted to "PLOS ONE." Based on the advice received, I have decided that your manuscript could be reconsidered for publication should you be prepared to incorporate minor revisions. When preparing your revised manuscript, you are asked to carefully consider the reviewers' comments, which can be found in the system submit a list of responses to the comments and mark them in red if possible.

Best regards

Mohammed S. Abdo

Academic Editor

Reviewers' comments:

Reviewer's Responses to Questions

**Comments to the Author**

1. Is the manuscript technically sound, and do the data support the conclusions?

Reviewer #1: Yes

Reviewer #2: Yes

2. Has the statistical analysis been performed appropriately and rigorously? 

Reviewer #1: N/A

Reviewer #2: N/A

3. Have the authors made all data underlying the findings in their manuscript fully available?

Reviewer #1: Yes

Reviewer #2: Yes

4. Is the manuscript presented in an intelligible fashion and written in standard English?

Reviewer #1: Yes

Reviewer #2: Yes

5. Review Comments to the Author

Reviewer #1: Strengths:

- The article addresses an important problem of solving the time-fractional gas dynamics equation using a novel analytical approach. Fractional calculus has become popular for modeling various physical phenomena, so solving fractional differential equations is an active research area.

- The authors provide a clear explanation of the methodology which combines the Elzaki transform, Caputo-Fabrizio fractional derivative and Adomian decomposition method. The step-by-step working is presented in a structured way.

- Two examples, one homogeneous and one non-homogeneous, are solved to demonstrate the applicability of the proposed technique. Numerical results and graphs are provided for different values of the fractional order β.

- Comparisons with exact solutions when β=1 show that the method yields accurate approximations. The results indicate the effect of the fractional order on the solution behavior.

- The article is well-written overall with adequate explanations of the mathematical concepts. The introduction provides motivation by describing applications of the gas dynamics equation.

Weaknesses:

- The article lacks some mathematical rigor in places. Some steps in the working, such as decomposing the solution into a series, are stated without justification.

- Convergence analysis of the approximation method is missing. Some discussion on error bounds would make the results stronger.

- More examples could be provided to establish the general applicability of the method for different types of gas dynamics equations.

- The physical interpretation of the effect of the fractional order on the solution behavior is not provided. This could give more insight into the results.

- There are a few typos and minor formatting issues in the article.

Recommendations:

- Add mathematical proofs or citations where appropriate to improve the rigor.

- Include some theoretical analysis of the accuracy and convergence of the approximation technique.

- Demonstrate the method on a wider variety of test problems involving different types of nonlinearities, boundary conditions etc.

- Provide physical explanations of the effect observed when changing the fractional order β.

- Expand the introduction to provide a more thorough literature review of existing methods for solving fractional gas dynamics equations.

- Carefully proofread the article to fix typos, formatting, and grammar issues.

- Overall, the paper presents a novel application of a useful analytical technique but needs some more mathematical depth and discussion of results to improve its potential for publication. Addressing the above points would strengthen the paper.

Reviewer #2: Dear Authors,

I would recommend publication of the manuscript in PLOS ONE provided that the authors can address the following issues properly.

1. Please check the Abstract and Conclusion in terms of the grammar English language for example the abstract change to (In this article, Elzaki decomposition method (EDM) has been applied to approximate the analytical solution of the time-fractional gas-dynamics equation. The time-fractional derivative is used in the Caputo-Fabrizio sense. The proposed method is implemented on homogenous and non-homogenous cases of the time-fractional gas-dynamics equation. A comparison between the exact and approximate solutions is also provided to show the validity and accuracy of the technique. A graphical representation of all the retrieved solutions is shown for different values of the fractional parameter. The time development of all solutions is also represented in 2D graphs. The obtained results may help understand the physical systems governed by the gas-dynamics equation.)

2. the results mentioned in this manuscript are confirmed with the review.

3. To complete the literature review, the following papers should be added to the references list:

• Communications in Theoretical Physics 74 (7), 075003.

• Advances in Difference Equations 331 (doi.org/10.1186/s13662-020-02789-5).

Good luck

6. PLOS authors have the option to publish the peer review history of their article (what does this mean?). If published, this will include your full peer review and any attached files.

Reviewer #1: No

Reviewer #2: No

---

## [Author Response · Author response to Decision Letter 0]

23 Nov 2023

Dear Professor,

Please see the attached response file.

Thanks

---

## [Decision Letter · Decision Letter 1]

28 Feb 2024

Solution of time-fractional gas dynamics equation using Elzaki decomposition method with Caputo-Fabrizio fractional derivative

PONE-D-23-34314R1

Dear author/s

We’re pleased to inform you that your manuscript has been judged scientifically suitable for publication and will be formally accepted for publication once it meets all outstanding technical requirements.

Kind regards,

Kottakkaran Sooppy Nisar

Academic Editor

PLOS ONE

Additional Editor Comments (optional):

Reviewers' comments:

Reviewer's Responses to Questions

**Comments to the Author**

1. If the authors have adequately addressed your comments raised in a previous round of review and you feel that this manuscript is now acceptable for publication, you may indicate that here to bypass the “Comments to the Author” section, enter your conflict of interest statement in the “Confidential to Editor” section, and submit your "Accept" recommendation.

Reviewer #1: All comments have been addressed

Reviewer #2: All comments have been addressed

2. Is the manuscript technically sound, and do the data support the conclusions?

Reviewer #1: Yes

Reviewer #2: Yes

3. Has the statistical analysis been performed appropriately and rigorously? 

Reviewer #1: N/A

Reviewer #2: I Don't Know

4. Have the authors made all data underlying the findings in their manuscript fully available?

Reviewer #1: Yes

Reviewer #2: Yes

5. Is the manuscript presented in an intelligible fashion and written in standard English?

Reviewer #1: Yes

Reviewer #2: Yes

6. Review Comments to the Author

Reviewer #1: Tha authors have responded. They have implemented all the suggestions and I my recommendations are position for the publication.

Reviewer #2: Dear Authors

Thank you very much for the correction of the article according to my opinion.

Wishes the best for you

7. PLOS authors have the option to publish the peer review history of their article (what does this mean?). If published, this will include your full peer review and any attached files.

Reviewer #1: No

Reviewer #2: No

---

## [Editor Report · Acceptance letter]

1 Apr 2024

PONE-D-23-34314R1 

PLOS ONE

Dear Dr. Emadifar, 

I'm pleased to inform you that your manuscript has been deemed suitable for publication in PLOS ONE. Congratulations! Your manuscript is now being handed over to our production team.

Kind regards, 

on behalf of

Prof. Kottakkaran Sooppy Nisar 

Academic Editor

PLOS ONE